# Automated scalable segmentation of neurons from multispectral images

**Uygar Sümbül**
Grossman Center for the Statistics of Mind
and Dept. of Statistics, Columbia University

**Douglas Roossien Jr.**
University of Michigan Medical School

**Fei Chen**
MIT Media Lab and McGovern Institute

**Nicholas Barry**
MIT Media Lab and McGovern Institute

**Edward S. Boyden**
MIT Media Lab and McGovern Institute

**Dawen Cai**
University of Michigan Medical School

**John P. Cunningham**
Grossman Center for the Statistics of Mind
and Dept. of Statistics, Columbia University

**Liam Paninski**
Grossman Center for the Statistics of Mind
and Dept. of Statistics, Columbia University

## Abstract

Reconstruction of neuroanatomy is a fundamental problem in neuroscience. Stochastic expression of colors in individual cells is a promising tool, although its use in the nervous system has been limited due to various sources of variability in expression. Moreover, the intermingled anatomy of neuronal trees is challenging for existing segmentation algorithms. Here, we propose a method to automate the segmentation of neurons in such (potentially pseudo-colored) images. The method uses spatio-color relations between the voxels, generates supervoxels to reduce the problem size by four orders of magnitude before the final segmentation, and is parallelizable over the supervoxels. To quantify performance and gain insight, we generate simulated images, where the noise level and characteristics, the density of expression, and the number of fluorophore types are variable. We also present segmentations of real Brainbow images of the mouse hippocampus, which reveal many of the dendritic segments.

## 1   Introduction

Studying the anatomy of individual neurons and the circuits they form is a classical approach to understanding how nervous systems function since Ramón y Cajal's founding work. Despite a century of research, the problem remains open due to a lack of technological tools: mapping neuronal structures requires a large field of view, a high resolution, a robust labeling technique, and computational methods to sort the data. Stochastic labeling methods have been developed to endow individual neurons with color tags [1, 2]. This approach to neural circuit mapping can utilize the light microscope, provides a high-throughput and the potential to monitor the circuits over time, and complements the dense, small scale connectomic studies using electron microscopy [3] with its large field-of-view. However, its use has been limited due to its reliance on manual segmentation.

The initial stochastic, spectral labeling (Brainbow) method had a number of limitations for neuroscience applications including incomplete filling of neuronal arbors, disproportionate expression of the nonrecombined fluorescent proteins in the transgene, suboptimal fluorescence intensity, and color shift during imaging. Many of these limitations have since improved [4] and developments

in various aspects of light microscopy provide further opportunities [5, 6, 7, 8]. Moreover, recent approaches promise a dramatic increase in the number of (pseudo) color sources [9, 10, 11]. Taken together, these advances have made light microscopy a much more powerful tool for neuroanatomy and connectomics. However, existing automated segmentation methods are inadequate due to the spatio-color nature of the problem, the size of the images, and the complicated anatomy of neuronal arbors. Scalable methods that take into account the high-dimensional nature of the problem are needed.

Here, we propose a series of operations to segment 3-D images of stochastically tagged nervous tissues. Fundamentally, the computational problem arises due to insufficient color consistency within individual cells, and the voxels occupied by more than one neuron. We denoise the image stack through collaborative filtering [12], and obtain a supervoxel representation that reduces the problem size by four orders of magnitude. We consider the segmentation of neurons as a graph segmentation problem [13], where the nodes are the supervoxels. Spatial discontinuities and color inhomogeneities within segmented neurons are penalized using this graph representation. While we concentrate on neuron segmentation in this paper, our method should be equally applicable to the segmentation of other cell classes such as glia.

To study various aspects of stochastic multispectral labeling, we present a basic simulation algorithm that starts from actual single neuron reconstructions. We apply our method on such simulated images of retinal ganglion cells, and on two different real Brainbow images of hippocampal neurons, where one dataset is obtained by expansion microscopy [5].

## 2    Methods

Successful segmentations of color-coded neural images should consider both the connected nature of neuronal anatomy and the color consistency of the Brainbow construct. However, the size and the noise level of the problem prohibit a voxel-level approach (Fig. 1). Methods that are popular in hyperspectral imaging applications, such as nonnegative matrix factorization [14], are not immediately suitable either because the number of color channels are too few and it is not easy to model neuronal anatomy within these frameworks. Therefore, we develop (i) a supervoxelization strategy, (ii) explicitly define graph representations on the set of supervoxels, and (iii) design the edge weights to capture the spatio-color relations (Fig. 2a).

### 2.1    Denoising the image stack

Voxel colors within a neurite can drift along the neurite, exhibit high frequency variations, and differ between the membrane and the cytoplasm when the expressed fluorescent protein is membrane-binding (Fig. 1). Collaborative filtering generates an extra dimension consisting of similar patches within the stack, and applies filtering in this extra dimension rather than the physical dimensions. We use the BM4D denoiser [12] on individual channels of the datasets, assuming that the noise is Gaussian. Figure 2 demonstrates that the boundaries are preserved in the denoised image.

### 2.2    Dimensionality reduction

We make two basic observations to reduce the size of the dataset: (i) Voxels expressing fluorescent proteins form the foreground, and the dark voxels form the much larger background in typical Brainbow settings. (ii) The basic promise of Brainbow suggests that nearby voxels within a neurite have very similar colors. Hence, after denoising, there must be many topologically connected voxel sets that also have consistent colors.

The watershed transform [15] considers its input as a topographic map and identifies regions associated with local minima ("catchment basins" in a flooding interpretation of the topographic map). It can be considered as a *minimum spanning forest* algorithm, and obtained in linear time with respect to the input size [16, 17]. For an image volume $V = V(x, y, z, c)$, we propose to calculate the topographical map $T$ (disaffinity map) as

$$T(x, y, z) = \max_{t \in \{x, y, z\}} \max_c |G_t(x, y, z, c)|, \tag{1}$$

where $x$, $y$, $z$ denote the spatial coordinates, $c$ denotes the color coordinate, and $G_x$, $G_y$, $G_z$ denote the spatial gradients of $V$ (nearest neighbor differencing). That is, *any* edge with significant deviation in *any* color channel will correspond to a "mountain" in the topographic map. A flooding parameter, $f$, assigns the local minima of $T$ to catchment basins, which partition $V$ together with the boundary voxels. We assign the boundaries to neighboring basins based on color proximity. The background is

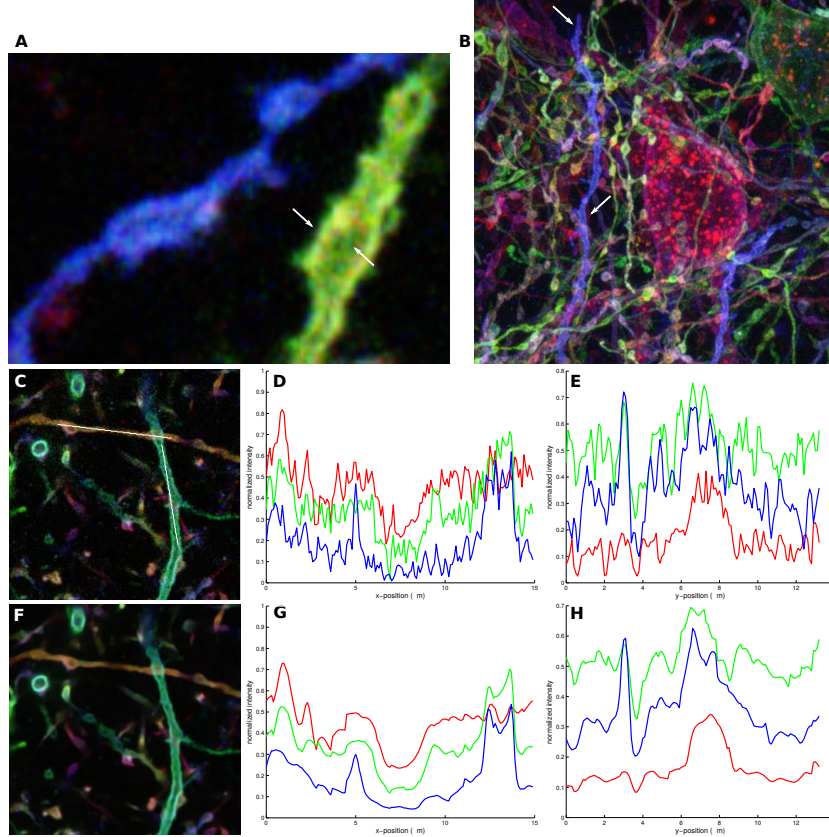

Figure 1: **Multiple noise sources affect the color consistency in Brainbow images. a,** An $85 \times 121$ Brainbow image patch from a single slice (physical size: $8.5\mu \times 12.1\mu$). Expression level differs significantly between the membrane and the cytoplasm along a neurite (arrows). **b,** A maximum intensity projection view of the 3-d image stack. Color shifts along a single neurite, which travels to the top edge and into the page (arrows). **c,** A $300 \times 300$ image patch from a single slice of a different Brainbow image (physical size: $30\mu \times 30\mu$). **d,** The intensity variations of the different color channels along the horizontal line in **c**. **e,** Same as **d** for the vertical line in **c**. **f,** The image patch in **c** after denoising. **g–h,** Same as **d** and **e** after denoising. For the plots, the range of individual color channels is $[0, 1]$.

the largest and darkest basin. We call the remaining objects supervoxels [18, 19]. Let $F$ denote the binary image identifying all of the foreground voxels.

Objects without interior voxels (e.g., single-voxel thick dendritic segments) may not be detected by Eq. 1 (Supp. Fig. 1). We recover such "bridges" using a topology-preserving warping (in this case, only shrinking is used.) of the thresholded image stack into $F$ [20, 21]:

$$B = \mathcal{W}(I_\theta, F), \tag{2}$$

where $I_\theta$ is binary and obtained by thresholding the intensity image at $\theta$. $\mathcal{W}$ returns a binary image $B$ such that $B$ has the same topology as $I_\theta$ and agrees with $F$ as much as possible. Each connected component of $B \wedge \bar{F}$ (foreground of $B$ and background of $F$) is added to a neighboring supervoxel based on color proximity, and discarded if no spatial neighbors exist (Supp. Text).

We ensure the color homogeneity within supervoxels by dividing non-homogeneous supervoxels (e.g., large color variation across voxels) into connected subcomponents based on color until the desired homogeneity is achieved (Supp. Text). We summarize each supervoxel's color by its mean color.

We apply local heuristics and spatio-color constraints iteratively to further reduce the data size and demix overlapping neurons in voxel space (Fig. 2f,g and Supp. Text). Supp. Text provides details on the parallelization and complexity of these steps and the method in general.

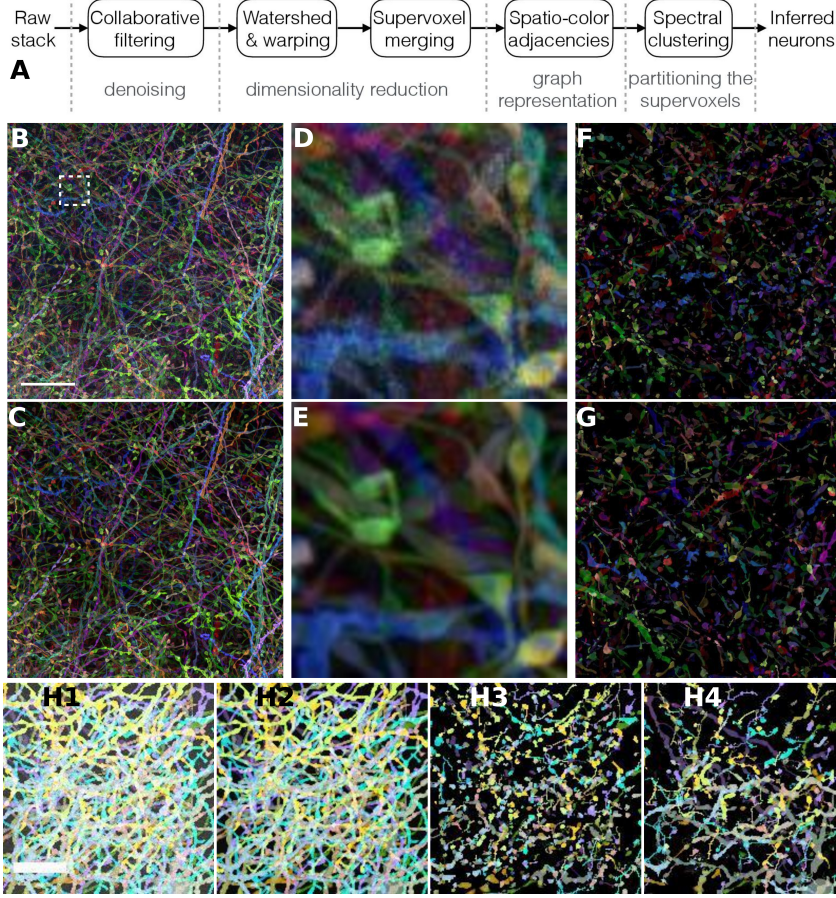

Figure 2: Best viewed digitally. **a,** A schematic of the processing steps **b,** Max. intensity projection of a raw Brainbow image **c,** Max. intensity projection of the denoised image **d,** A zoomed-in version of the patch indicated by the dashed square in **b**. **e,** The corresponding denoised image. **f,** One-third of the supervoxels in the top-left quadrant (randomly chosen). **g,** Same as **f** after the merging step. **h1-h4,** Same as **b,c,f,g** for simulated data. Scale bars, $20\mu m$.

## 2.3 Clustering the supervoxel set

We consider the supervoxels as the nodes of a graph and express their spatio-color similarities through the existence (and the strength) of the edges connecting them, summarized by a highly sparse adjacency matrix. Removing edges between supervoxels that aren't spatio-color neighbors avoids spurious links. However, this procedure also removes many genuine links due to high color variability (Fig. 1). Moreover, it cannot identify disconnected segments of the same neuron (e.g., due to limited field-of-view). Instead, we adjust the spatio-color neighborhoods based on the "reliability" of the colors of the supervoxels. Let $S$ denote the set of supervoxels in the dataset. We define the sets of reliable and unreliable supervoxels as $S_r = \{s \in S : n(s) > t_s, h(s) < t_d\}$ and $S_u = S \setminus S_r$, respectively, where $n(s)$ denotes the number of voxels in $s$, $h(s)$ is a measure of the color heterogeneity (e.g., the maximum difference between intensities across all color channels), $t_s$ and $t_d$ are the corresponding thresholds.

We describe a graph $G = (V, E)$, where $V$ denotes the vertex set (supervoxels) and $E = E_s \cup E_c \cup E_{\bar{s}}$ denotes the edges between them:

$$E_s = \{(ij) : \delta_{ij} < \epsilon_s, i \neq j\}$$
$$E_c = \{(ij) : s_i, s_j \in S_r, d_{ij} < \epsilon_c, i \neq j\}$$
$$E_{\bar{s}} = \{(ij), (ji) : s_i \in S_u, (ij) \notin E_s, O_i(j) < k_{\min} - K_i, i \neq j\}, \qquad (3)$$

where $\delta_{ij}$, $d_{ij}$ are the spatial and color distances between $s_i$ and $s_j$, respectively. $\epsilon_s$ and $\epsilon_c$ are the corresponding maximum distances. An unreliable supervoxel with too few spatial neighbors is allowed to have up to $k_{\min}$ edges via proximity in color space. Here, $O_i(j)$ is the order of supervoxel $s_j$ in terms of the color distance from supervoxel $s_i$, and $K_i$ is the number of $\epsilon_s$-spatial neighbors of $s_i$. (Note the symmetric formulation in $E_{\bar{s}}$.) Then, we construct the adjacency matrix as

$$A(i, j) = \begin{cases} e^{-\alpha d_{ij}^2}, & (ij) \in E \\ 0, & \text{otherwise} \end{cases} \qquad (4)$$

where $\alpha$ controls the decay in affinity with respect to distance in color. We use k-d tree structures to efficiently retrieve the color neighborhoods [22]. Here, the distance between two supervoxels is $\min_{v\in V, u\in U} D(v, u)$, where $V$ and $U$ are the voxel sets of the two supervoxels and $D(v, u)$ is the Euclidean distance between voxels $v$ and $u$.

A classical way of partitioning graph nodes that are nonlinearly separable is by minimizing a function (e.g., the sum or the maximum) of the edge weights that are severed during the partitioning [23]. Here, we use the normalized cuts algorithm [24, 13] with two simple modifications: the $k$-means step is weighted by the sizes of the supervoxels and initialized by a few iterations of $k$-means clustering of the supervoxel colors only (Supp. Text). The resulting clusters partition the image stack (together with the background), and represent a segmentation of the individual neurons within the image stack. An estimate of the number of neurons can be obtained from a Dirichlet process mixture model [25]. While this estimate is often rough [26], the segmentation accuracy appears resilient to imperfect estimates (Fig. 4c).

### 2.4 Simulating Brainbow tissues

We create basic simulated Brainbow image stacks from volumetric reconstructions of single neurons (Algorithm 1). For simplicity, we model the neuron color shifts by a Brownian noise component on the tree, and the background intensity by a white Gaussian noise component (Supp. Text).

We quantify the segmentation quality of the voxels using the adjusted Rand index (ARI), whose maximum value is 1 (perfect agreement), and expected value is 0 for random clusters [27]. (Supp. Text)

---

**Algorithm 1** Brainbow image stack simulation

---

**Require:** number of color channels $C$, set of neural shapes $S = \{n_i\}_i$, stack (empty, 3d space + color), background noise variability $\sigma_1$, neural color variability $\sigma_2$, $r$, saturation level $M$
1: **for** $n_i \in S$ **do**
2:     Shift and rotate neuron $n_i$ to minimize overlap with existing neurons in the stack
3:     Generate a uniformly random color vector $v_i$ of length $C$
4:     Identify the connected components of $c_{ij}$ of $n_i$ within the stack
5:     **for** $c_{ij} \in \{c_{ij}\}_j$ **do**
6:         Pre-assign $v_i$ to $r\%$ of the voxels of $c_{ij}$
7:         $C$-dimensional random walk on $c_{ij}$ with steps $\mathcal{N}(0, \sigma_1^2 \mathbf{I})$ (Supp. Text)
8:     **end for**
9:     Add neuron $n_i$ to the stack (with additive colors for shared voxels)
10: **end for**
11: Add white noise to each voxel generated by $\mathcal{N}(0, \sigma_2^2 \mathbf{I})$
12: **if** brightness exceeds $M$ **then**
13:     Saturate at $M$
14: **end if**
15: **return** stack

---

## 3  Datasets

To simulate Brainbow image stacks, we used volumetric single neuron reconstructions of mouse retinal ganglion cells in Algorithm 1. The dataset is obtained from previously published studies [28, 29]. Briefly, the voxel size of the images is $0.4\mu \times 0.4\mu \times 0.5\mu$, and the field of view of individual stacks is $320\mu \times 320\mu \times 70\mu$ or larger. We evaluate the effects of different conditions on a central portion of the simulated image stack.

Both real datasets are images of the mouse hippocampal tissue. The first dataset has $1020 \times 1020 \times 225$ voxels (voxel size: $0.1 \times 0.1 \times 0.3\mu^3$), and the tissue was imaged at 4 different frequencies (channels). The second dataset has $1080 \times 1280 \times 134$ voxels with an effective voxel size of $70 \times 70 \times 40nm$, where the tissue was $4\times$ linearly expanded [5], and imaged at 3 different channels. The Brainbow constructs were delivered virally, and approximately $5\%$ of the neurons express a fluorescence gene.

## 4  Results

Parameters used in the experiments are reported in Supp. Text.

Fig. 1b, d, and e depict the variability of color within individual neurites in a single slice and through the imaging plane. Together, they demonstrate that the voxel colors of even a small segment of a

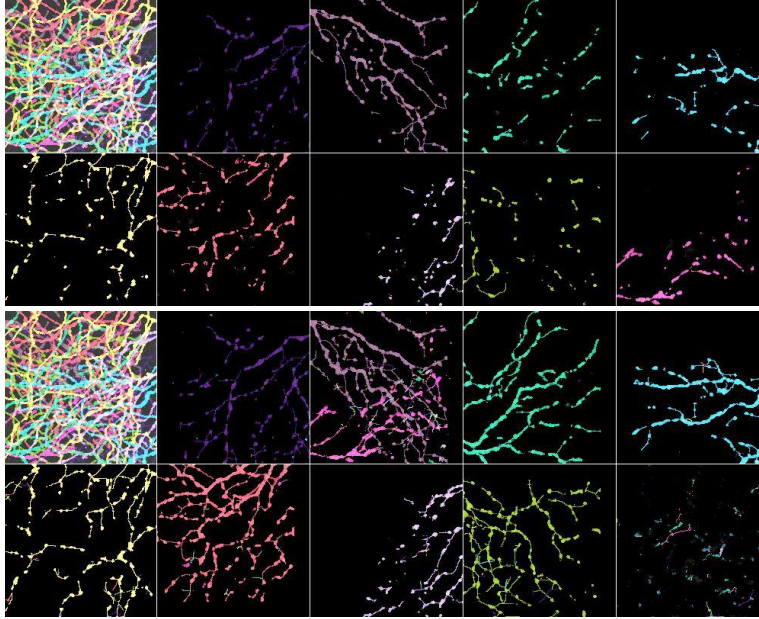

Figure 3: **Segmentation of a simulated Brainbow image stack.** Adjusted Rand index of the foreground is $0.80$. **Pseudocolor representation of 4-channel data. Top:** maximum intensity projection of the ground truth. Only the supervoxels that are occupied by a single neuron are shown. **Bottom:** maximum intensity projection of the reconstruction. The top-left corners show the whole image stack. All other panels show the maximum intensity projections of the supervoxels assigned to a single cluster (inferred neuron).

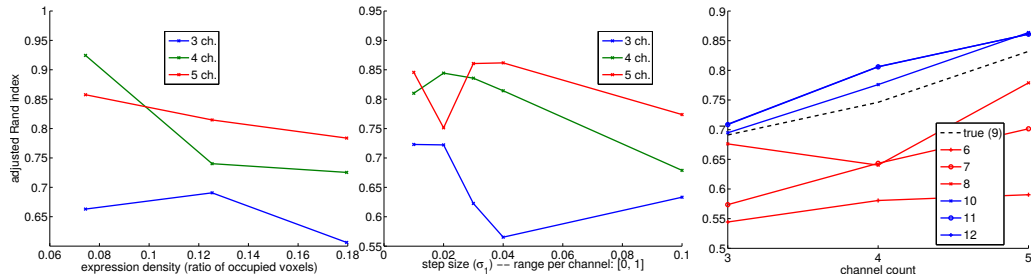

Figure 4: **Segmentation accuracy of simulated data a,** Expression density (ratio of voxels occupied by at least one neuron) vs. ARI. **b,** $\sigma_1$ (Algorithm 1) vs. ARI. **c,** Channel count vs. ARI for a 9-neuron simulation, where $K \in [6, 12]$. ARI is calculated for the foreground voxels. See Supp. Fig. 7 for ARI values for all voxels.

neuron's arbor can occupy a significant portion of the dynamic range in color with the state-of-the-art Brainbow data. Fig. 1c-e show that collaborative denoising removes much of this noise while preserving the edges, which is crucial for segmentation. Fig. 2b-e and h demonstrate a similar effect on a larger scale with real and simulated Brainbow images.

Fig. 2 shows the raw and denoised versions of the $1020 \times 1020 \times 225$ image, and a randomly chosen subset of its supervoxels (one-third). The original set had $6.2 \times 10^4$ supervoxels, and the merging routine decreased this number to $3.9 \times 10^4$. The individual supervoxels grew in size while avoiding mergers with supervoxels of different neurons. This set of supervoxels, together with a (sparse) spatial connectivity matrix, characterizes the image stack. Similar reductions are obtained for all the real and simulated datasets.

Fig. 3 shows the segmentation of a simulated $200 \times 200 \times 100$ (physical size: $80\mu \times 80\mu \times 50\mu$) image patch. (Supp. Fig. 2 shows all three projections, and Supp. Fig. 3 shows the density plot through the $z$-axis.) In this particular example, the number of neurons within the image is 9, $\sigma_1 = 0.04$, $\sigma_2 = 0.1$, and the simulated tissue is imaged using 4 independent channels. Supp. Fig. 4 shows a patch from a single slice to visualize the amount of noise. The segmentation has an adjusted Rand index of $0.80$ when calculated for the detected foreground voxels, and $0.73$ when calculated for all voxels. (In some cases, the value based on all voxels is higher.) The ground truth image displays only those supervoxels all of whose voxels belong to a single neuron. The bottom part of Fig. 3 shows

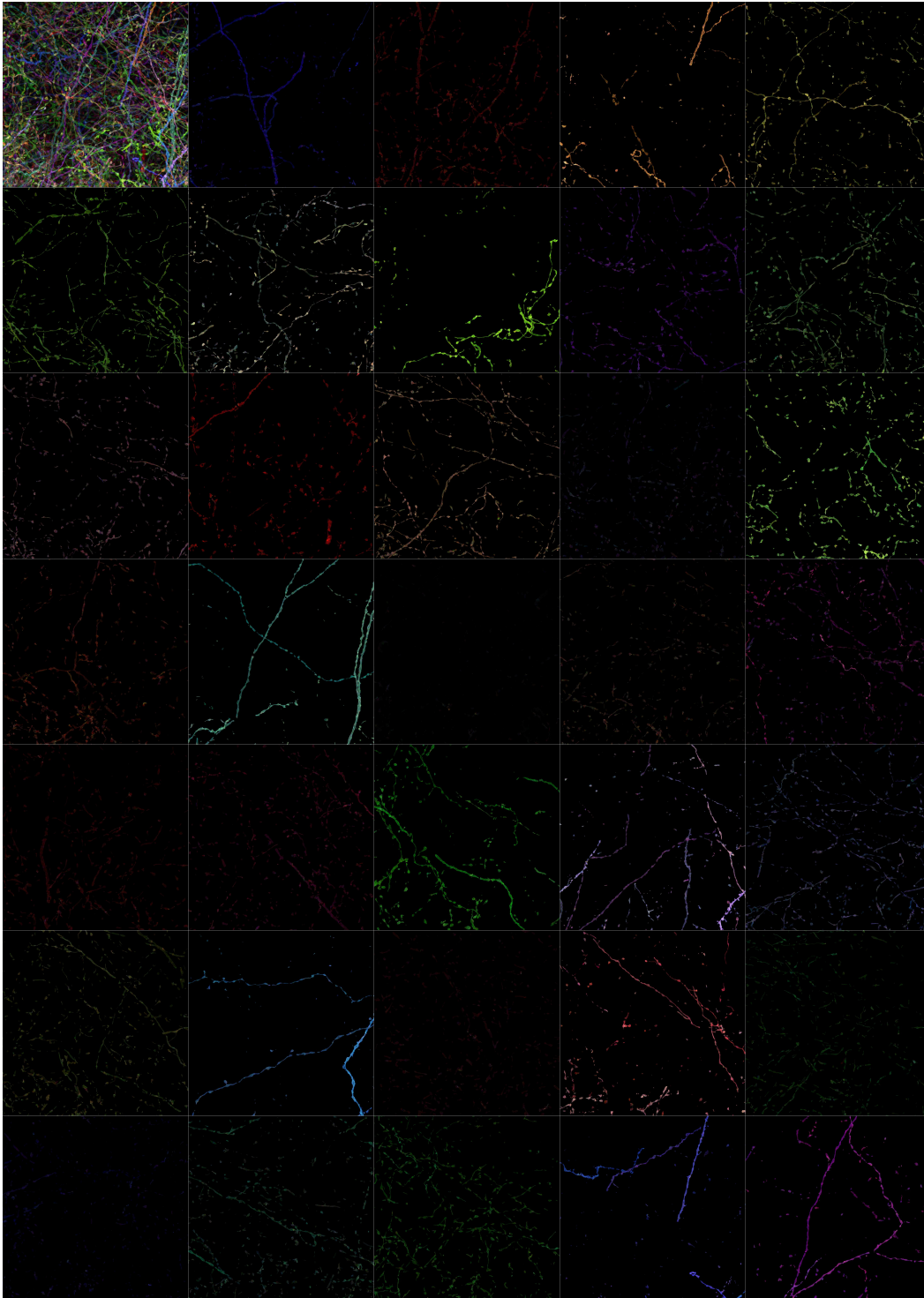

Figure 5: **Segmentation of a Brainbow stack – best viewed digitally. Pseudo-color representation of 4-channel data.** The physical size of the stack is $102\mu \times 102\mu \times 68\mu$. The top-left corner shows the maximum intensity projection of the whole image stack, all other panels show the maximum intensity projections of the supervoxels assigned to a single cluster (inferred neuron).

that many of these supervoxels are correctly clustered to preserve the connectivity of neuronal arbors. There are two important mistakes in clusters 4 (merger) and 9 (spurious cluster). These are caused by aggressive merging of supervoxels (Supp. Fig. 5), and the segmentation quality improves with the inclusion of an extra imaging channel and more conservative merging (Supp. Fig. 6). We plot the performance of our method under different conditions in Fig. 4 (and Supp. Fig. 7). We set the noise standard deviation to $\sigma_1$ in the denoiser, and ignored the contribution of $\sigma_2$. Increasing the number of observation channels improves the segmentation performance. The clustering accuracy degrades gradually with increasing neuron-color noise ($\sigma_1$) in the reported range (Fig. 4b). The accuracy does not seem to degrade when the cluster count is mildly overestimated, while it decays quickly when the count is underestimated (Fig. 4c).

Fig. 5 displays the segmentation of the $1020 \times 1020 \times 225$ image. While some mistakes can be spotted by eye, most of the neurites can be identified and simple tracing tools can be used to obtain final skeletons/segmentations [30, 31]. In particular, the identified clusters exhibit homogeneous colors and dendritic pieces that either form connected components or miss small pieces that do not preclude the use of those tracing tools. Some clusters appear empty while a few others seem to comprise segments from more than one neuron, in line with the simulation image (Fig. 2.4).

Supp. Fig. 8 displays the segmentation of the $4\times$ expanded, $1080 \times 1280 \times 134$ image. While the two real datasets have different characteristics and voxel sizes, we used essentially the same parameters for both of them throughout denoising, supervoxelization, merging, and clustering (Supp. Text). Similar to Fig. 5, many of the processes can be identified easily. On the other hand, Supp. Fig. 8 appears more fragmented, which can be explained by the smaller number of color channels (Fig. 4).

## 5 Discussion

Tagging individual cells with (pseudo)colors stochastically is an important tool in biological sciences. The versatility of genetic tools for tagging synapses or cell types and the large field-of-view of light microscopy positions multispectral labeling as a complementary approach to electron microscopy based, small-scale, dense reconstructions [3]. However, its use in neuroscience has been limited due to various sources of variability in expression. Here, we demonstrate that automated segmentation of neurons in such image stacks is possible. Our approach considers both accuracy and scalability as design goals.

The basic simulation proposed here (Algo. 1) captures the key aspects of the problem and may guide the relevant genetics research. Yet, more detailed biophysical simulations represent a valuable direction for future work. Our simulations suggest that the segmentation accuracy increases significantly with the inclusion of additional color channels, which coincides with ongoing experimental efforts [9, 10, 11]. We also note that color constancy of individual neurons plays an important role both in the accuracy of the segmentation (Fig. 4) and the supervoxelized problem size.

While we did not focus on post-processing in this paper, basic algorithms (e.g., reassignment of small, isolated supervoxels) may improve both the visualization and the segmentation quality. Similarly, more elaborate formulations of the adjacency relationship between supervoxels can increase the accuracy. Finally, supervised learning of this relationship (when labeled data is present) is a promising direction, and our methods can significantly accelerate the generation of training sets.

## 6 Acknowledgments

The authors thank Suraj Keshri and Min-hwan Oh (Columbia University) for useful conversations.

Funding for this research was provided by ARO MURI W911NF-12-1-0594, DARPA N66001-15-C-4032 (SIMPLEX), and a Google Faculty Research award; in addition, this work was supported by the Intelligence Advanced Research Projects Activity (IARPA) via Department of Interior/ Interior Business Center (DoI/IBC) contract number D16PC00008. The U.S. Government is authorized to reproduce and distribute reprints for Governmental purposes notwithstanding any copyright annotation thereon. Disclaimer: The views and conclusions contained herein are those of the authors and should not be interpreted as necessarily representing the official policies or endorsements, either expressed or implied, of IARPA, DoI/IBC, or the U.S. Government.

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
