[Supplementary Material]

# Supplementary information: Automated scalable segmentation of neurons from multispectral images

## Supplementary Text

**Parameter choices and feature representations**

Similar results are obtained around a neighborhood of these suggested values. Unless otherwise noted, the same parameter values were used for all three reported experiments: two real hippocampal datasets acquired by different labs and under different conditions, and a set of simulated retinal datasets with different parameters. The subsection of the main text that refers to these parameters are indicated in square brackets.

**Color features:** For every color triplet, we obtain the L-u-v representation (Schanda, 2007), and calculate the top $C$ principal components of the concatenated L-u-v representations, where $C$ is the number of color channels. If the data has more than 3 channels, for affinity calculations between neighboring supervoxels, we normalize the colors before the L-u-v transformation. [Dimensionality reduction]

**Supervoxel reliability:** $t_s = 50$, $t_d = 0.5$ (before L-u-v transformation) [Clustering the supervoxel set]

**Edge set parameters:** $\epsilon_s = \sqrt{3}$ (26-neighborhood for isotropic data), $\epsilon_c = 20 \times \sqrt{C/4}$ (by inspecting typical color radius within individual neurons and manual adjustment), $k_{\min} = 5$. [Clustering the supervoxel set]

**Edge strength decay:** $\alpha = 2 \times 10^{-3}$ (by inspecting typical color radius and manual adjustment) [Clustering the supervoxel set]

**Flooding parameter for watershed:** $f = 0.01$ with 26-neighborhood. (This affects computation time more than quality because subdividing via the maximum color perimeter can catch inhomogeneous supervoxels.) [Dimensionality reduction]

**Maximum color perimeter for supervoxel homogeneity:** (Supp. Algo. 1) $p = 0.5$ for each channel when the intensity is in $[0, 1]$ (by inspecting data – see Fig. 1). [Dimensionality reduction]

**Image thresholding for warping:** $\theta = 0.1 \times \sqrt{C/4}$ before L-u-v transformation (for the expansion microscopy data, $\theta = 0.2$) [Dimensionality reduction]

**Noise standard deviation for denoising:** $\sigma = 1/8$ when the intensity is in $[0, 1]$ for individual channels. [Denoising the image stack]

**Cluster (neuron) counts:** For the dataset in Fig. 5, the mixture model (Kurihara et al., 2007) suggested 52 clusters based on the colors of the supervoxels. The same routine returned 29 clusters when run on $\frac{1}{5}$ of the supervoxels. We chose $K = 34$ for a compact presentation. For the dataset in

Supp. Fig. 8, we used $K = 19$, which is what the mixture model suggested based on the colors of the supervoxels. [Clustering the supervoxel set]

## Spatial distance calculation

The spatial distance between two supervoxels is calculated as $\min_{v \in V_1, u \in V_2} D(v, u)$, where $V_1$ and $V_2$ are the voxel sets of the two supervoxels and $D(v, u)$ is the Euclidean distance between voxels $v$ and $u$. Only the boundary voxels need to be considered, and extremal values in each coordinate are used to identify many supervoxel pairs farther than $\epsilon_s$ without exact calculation over the voxels. Only the spatial distances between nearby supervoxels need to be computed.

## Color-based subdivision of supervoxels

Let the $n \times C$ matrix $V_i$ denote the colors of all $n$ voxels of the supervoxel $s_i$. Supp. Algo. 1 divides the supervoxels into smaller supervoxels until the desired homogeneity is achieved.

---

**Supplementary Algorithm 1** Subdivide supervoxels

---

**Require:** $S = \{s_i\}_i$ (set of supervoxels), $\{V_i\}_i$, $p_{\max}$ (threshold)
1: $S_{\text{new}} = \{\}$
2: **for** $s_i \in S$ **do**
3:      $p = \max_{c \in C} \max(V_i(:, c)) - \min(V_i(:, c))$
4:      **if** $p < p_{\max}$ **then**
5:          Add $s_i$ to $S_{\text{new}}$
6:      **else**
7:          Divide the voxels into 2 sets $T_1$ and $T_2$ based on color (e.g., using $k$-means, hierarchical clustering, etc.)
8:          Add the connected components of $T_1$ and $T_2$ to $S$
9:      **end if**
10:    Remove $s_i$ from $S$
11: **end for**
12: **return** $S_{\text{new}}$

---

## Simulation data

RGC arbors stratify in the retina, distributing their dendritic length within a slab. To achieve denser simulations, we did not shift the neurons much in $z$ (The density numbers calculated in Fig. 4 are obtained by considering the $[35\mu m, 50\mu m]$ region in Supp. Fig. 3.) We obtain simulated stacks by varying the expression density ($|S| \in \{5, 9, 13\}$), the channel count ($C \in \{3, 4, 5\}$), the neuron color consistency ($\sigma_1 \in \{0.01, 0.02, 0.03, 0.04, 0.1\}$), and the background noise ($\sigma_2 \in \{0.05, 0.1\}$).

The random walk on a connected component assigns a color to a voxel by (i) calculating the mean color of the neighboring voxels that were previously visited, and (ii) adding the random noise step to this mean value.

## Adjusted Rand index

We quantify the segmentation quality of the voxels of the simulated dataset using the adjusted Rand index. The Rand index is a measure of the element pairs on which two partitions $P$ and $\hat{P}$ of the same set with $N$ elements agree: $R(P, \hat{P}) = 1 - \binom{N}{2} \sum_{i<j} |\delta(l_i, l_j) - \delta(\hat{l}_i, \hat{l}_j)|$, where $l_i$ ($\hat{l}_i$) denotes the label of element $i$ according to $P$ ($\hat{P}$), and $\delta$ is the indicator function. ($\delta(l_i, l_j) = 1$ if $l_i = l_j$, $\delta(l_i, l_j) = 0$ otherwise.) The adjusted Rand index corrects for chance, has a more sensitive dynamic

range, and is defined as $A = (R - E)/(M - E)$, where $E$ is the expected value of the index and $M$ is the maximum value of the index, based on the number of elements in individual segments. Its maximum value is 1 (perfect agreement), and the expected value of the index is 0 for random clusters.

For the foreground based calculation, only the voxels that are assigned to the foreground after the watershed transform and warping are considered. For the image based calculation, all voxels are considered and the background is treated as a separate object.

**Merging supervoxels**

We apply local heuristics and spatio-color constraints iteratively to further reduce the data size and demix overlapping neurons in voxel space (Fig. 2): (i) supervoxels occupied by more than one neuron are detected and demixed by monitoring the improvement in non-negative least squares fit quality. (Supp. Algo 2.) (ii) neighboring supervoxels with similar colors and orientations, supervoxels with single spatial neighbors, and supervoxels all of whose neighbors have similar colors are merged. (iii) supervoxels that are spatial neighbors and that are assigned to the same cluster by an overclustering color $k$-means routine are merged. We implement (iii) to run in parallel over subgraphs of the full graph for scalability. A rough estimate of the number of neurons required by the oversegmentation routine is obtained by a Dirichlet process mixture model (Kurihara et al., 2007). The $k$-means algorithm uses a multiple of this rough estimate. Note that only a rough estimate (Miller and Harrison, 2013) is needed because of oversegmentation (Supp Algo. 3). This algorithm can be implemented to run in parallel over subgraphs of the full dataset.

---

**Supplementary Algorithm 2** Demixing of supervoxels

---

**Require:** $S = \{s_i\}_i$, $V$ (matrix of normalized supervoxel colors), $A$ (spatial affinity matrix), $M$ (maximum size), $\Delta$ (maximum color distance), $f$ (improvement factor)

1: **for** $s_i \in S$ **do**
2:     **if** $s_i$ has less than $M$ voxels **then**
3:         Retain the neighbors that have neighbors with color distance less than $\Delta$
4:         **if** $s_i$ has more than one spatial neighbors **then**
5:             **if** the minimum color distance between $s_i$ and its neighbors is larger than $\Delta$ **then**
6:                 Initialize $r = ||V(i,:)||_2^2$, $P = (0,0)$
7:                 **for** each neighbor pair $(i_1, i_2)$ **do**
8:                     $t = \min_x ||V([i_1, i_2],:)x - V(i,:)||_2^2$ subject to $x \geq 0$
9:                     **if** $t < r$ **then**
10:                         $r = t$, $P = (i_1, i_2)$
11:                     **end if**
12:                 **end for**
13:                 **if** $r < (\Delta/f)^2$ **then**
14:                     Assign the voxels of $s_i$ to both of $s_{P(1)}$ and $s_{P(2)}$
15:                     Update the spatial affinities of $s_{P(1)}$ and $s_{P(2)}$ in $A$ accordingly
16:                     Remove $s_i$ from $S$, $V$, and $A$
17:                 **end if**
18:             **end if**
19:         **end if**
20:     **end if**
21: **end for**
22: **return** $S$, $V$, $A$

---

**Supplementary Algorithm 3** Spatio-color merging of supervoxels

---

**Require:** $S = \{s_i\}_i$, (set of supervoxels) $K$ (rough estimate of the number of clusters), $k$ (oversegmentation factor)

1: $S_{\text{new}} = \{\}$
2: Divide $S$ into $kK$ clusters based on the colors of the supervoxels, using $k$-means
3: **for** $\kappa_1 \in \{1, \dots, kK\}$ **do**
4:     Find the connected components within the cluster $\kappa_1$
5:     Merge the supervoxels within the connected components of that cluster, and add to $S_{\text{new}}$
6: **end for**
7: **return** $S_{\text{new}}$

---

Supplementary Figure 1: **Top:** Maximum intensity projection of a raw Brainbow image. **Bottom left:** Foreground after watershed transform. Arrows point to six different thin dendritic pieces ("bridges") that were missed. **Bottom right:** Foreground after warping correction. Scale bar, $30\mu m$

Supplementary Figure 2: **The $z$ (top left), $x$ (top right), and $y$ (bottom left) maximum intensity projections of the raw simulation image shown in Fig. 3. Adjusted Rand index of the segmentation is** $0.80$**.**

Supplementary Figure 3: **The $z$-profile of the ground truth simulation image with** 13 **neurons, showing that a region is preferentially occupied. The range** $[0\mu m, 50\mu m]$ **corresponds to slices** 1 **to** 100 **so that most of the neuronal arbors are between slices** 70 **and** 100**.**

Supplementary Figure 4: **A** $60 \times 60$ **patch from a single slice (slice 90) of the simulation image shown in Fig. 3. Top:** raw. **Bottom:** denoised. Physical size: $15\mu m \times 15\mu m$

Supplementary Figure 5: **Aggressive merging generates supervoxels with inconsistent colors. Top:** Cluster 4 in the bottom part of Fig. 3 of the main text. **Bottom:** Cluster 9 in the bottom part of Fig. 3 of the main text. Close inspection reveals that some of the multi-colored regions comprise single supervoxels.

Supplementary Figure 6: **Segmentation of a simulated Brainbow image stack. Adjusted Rand index of the foreground is** $0.87$. **Pseudo-color representation of 5-channel data with more conservative supervoxel merging compared to Fig. 3 of the main text.** Maximum intensity projection of the segmentation. The top-left corner shows the whole image stack. All other panels show the maximum intensity projections of the supervoxels assigned to a single cluster (inferred neuron).

Supplementary Figure 7: **Segmentation accuracy of simulated data a,** Expression density (ratio of voxels occupied by at least one neuron) vs. ARI. **b,** $\sigma_1$ vs. ARI. **c,** Channel count vs. ARI for a 9-neuron simulation, where $K \in [6, 12]$. ARI is calculated for all voxels.

Supplementary Figure 8: **Segmentation of a** $4\times$ **linearly expanded Brainbow stack – best viewed digitally.** The physical size of the stack is $90\mu \times 76\mu \times 5\mu$. The top-left corner shows the maximum intensity projection of the whole image stack, all other panels show the maximum intensity projections of the supervoxels assigned to a single cluster (inferred neuron).

**Parallelization and complexity**

The denoising step (collaborative filtering) parallelizes over substacks (voxels) because the extra dimension is formed by local patches.

The watershed algorithm can also be parallelized over substacks. Moreover, it has a linear run-time with respect to the input size (Main Text).

Similarly, warping (shrinking) can be applied on those substacks. Querying the boundary voxels for flipping at each stage, and ordering them by brightness result in a fast implementation. It has a linear run-time with respect to the input size. (Simple voxel query can be performed over $3 \times 3 \times 3$ patches – See references in main text.)

Supervoxel merging can be performed in parallel on individual substacks. Merging subroutines use local rules to make local merges except for the spatio-color merging step, which uses the $k$-means algorithm.

Similarity calculations require extracting spatial and color neighborhoods of the supervoxels. Spatial distance calculation is discussed above. These value are precalculated. As mentioned in the main text, k-d tree structures are used to retrieve the color neighborhoods efficiently.

Finally, clustering is performed by the normalized cuts algorithm. We use an implicitly restarted block Lanczos method for computing the first few eigenvectors (Baglama et al., 2003).