[Reviews · NeurIPS 2016]

Reviewer 1

Summary

This is a highly interesting paper that studies the segmentation of neural tissue from genetically tagged light microscopy. It shows a wealth of analysis methods that are pipelined to help denoise and connect relevant pieces for retrieving single neurons. The techniques are a specific combination of computer vision pattern recognition and ML tools, showing very encouraging results on a very hard scientific problem. The paper shows a challenging application to the NIPS community but it is a bit unclear to the reviewer what can be learned from the pipeline beyond the very application. Also it is unclear what particular step in the process is essential to get right in order to achieve the impressive results. Concluding, an excellent scientific paper with impressive results not targeted well to the nips audience. It would be nice to discuss and understand the ML challenges at the poster.

Qualitative Assessment

an excellent scientific paper with impressive results not targeted well to the nips audience. It would be nice to discuss and understand the ML challenges at the final paper.

Confidence in this Review

3-Expert (read the paper in detail, know the area, quite certain of my opinion)


Reviewer 2

Summary

This paper describes a pipeline for segmenting neural structures from multi-channel microscopy images taken with a confocal microscope. A procedure is described for generating simulated data and the algorithm is benchmarked while varying key parameters of the simulation. Finally, the algorithm is applied on real data and the results are shown

Qualitative Assessment

"After reading the other reviews and rebuttal, I will increase my novelty score to 2. I had previously scored it based on the pipeline itself, but the application to Brainbow data seems novel and potentially interesting." This paper describes some very interesting data, which could potentially be used to segment and track neurons in microscopy images. This could be used to find connections between neurons, an active area of research with potentially high-impact in neuroscience. While the data looks great and promising, the pipeline follows a "classical" multi-step approach. A sequence of reasonable, but somewhat arbitrary steps are described, and many parameter choices are not motivated. It is doubtful then that this algorithm could generalize to even slightly different data, and it might require careful parameter tuning for each new field of view. It is suggested that the pipeline needs to be fast, hence why only basic algorithms are described. The authors argue that the dataset size is very large, but at 1020 x 1020 x 225 dimensions, I would argue that is actually quite small. High-resolution EM images are orders of magnitudes larger in size, and yet some principles algorithms for processing these do exist. Even a straightforward deep CNN classifier would be more robust, and much easier to implement too. That could be used in conjunction with the simulated datasets the authors describe, to provide labelled data.

Confidence in this Review

2-Confident (read it all; understood it all reasonably well)


Reviewer 3

Summary

The paper studies segmentation of neurons from multispectral images. The authors take 3D pseudocolor images obtained from stochastic tagging of nervous tissue in mouse hippocampus. They denoise the images using collaborative filtering, they use watershed transform to identify supervoxels, define distance between supervoxels and find the neurons in the resulting graph on supervoxels by normalized cuts algorithm. Because of the lack of ground truth data, the authors propose a method to create synthetic data. They evaluate their methods first on these simulated images, and second they segment actual neuron images and show segmentation results.

Qualitative Assessment

The paper is written in a clear understandable way except for some details (eg. authors say they adapt normalized cuts but do not specify how) and the results section which would need more work (the actual text describing the results does not go beyond describing the figures). The evaluation of, how good/useful is the method, is lacking - this could have been either some reasonable baseline method, comparison to a high quality segmentation from electron microscopy, or an estimate of how much human post-processing would be still required. Questions: - why aren't results using 4 channels better than using 3, but they are using 5 channels? Can you at least speculate? (Fig. 4a,b) - in the discussion section the authors claim the method is scalable, I would like to see at least runtimes that would backup such a claim. - how were the parameters of the method fixed (alpha, beta)? - the authors claim the method is robust to deviations from the color-consistency assumption (Fig. 4b) - how does the range used for sigma_1 relate to actual noise in real images? It should be possible to estimate sigma_1 for individual neurons segmented from the real data. - in the text (Sec. 2.3) the authors refer to evaluation of the method with respect to (in)correct estimate of the neuron count but this data is missing from the figure Minor comments: - comparing O(10) with O(10^8) is plain wrong - Fig. 1 the horizontal/vertical lines are nearly invisible - not speaking about a printout - Figs 1, 4 - too small illegible font - there are typos, please use spellchecker

Confidence in this Review

3-Expert (read the paper in detail, know the area, quite certain of my opinion)


Reviewer 4

Summary

The main confusion lies in whether simulating the brain bow stack is original or the supervoxel merging. The algorithm is also evaluated only on one dataset. It's unclear if this raises a bias or a highly over fittest model. No comparative study is made between standard models/ methods or any other standard dataset. Also in the denoising section the noise is considered gaussian, why is that?

Qualitative Assessment

The main confusion lies in whether simulating the brain bow stack is original or the supervoxel merging. The algorithm is also evaluated only on one dataset. It's unclear if this raises a bias or a highly over fittest model. No comparative study is made between standard models/ methods or any other standard dataset. Also in the denoising section the noise is considered gaussian, why is that? This paper looks like a collection of several standard algorithms to solve a problem. While the application can be very useful, it seems the paper fails to explain what is it's original contribution.

Confidence in this Review

2-Confident (read it all; understood it all reasonably well)


Reviewer 5

Summary

The authors proposed a methdo to automatically segment neurons from multispectral images. The algorithm is based on a 5 steps pipeline which makes us of standard techniques from image processing (voxelization, watershed) and from machine learning (spectral clustering) in order to reconstruct the neurons shape.

Qualitative Assessment

After reading the author's rebuttal I have increased the technical quality to 2 and after reading the the other reviews I increased the potential impact to 3. The authors replied to many questions but not to all, in particular the answer was not satisfactory to the question about the parameter K which is one of the crucial parameter in any segmentation algorithm. Why they did not provide the results using the suggested automatic method in Fig4 instead of cyclying on possible (wrong) number of clusters ? this will asses the generality of the method. I would have expected to see in the results the performances with at least one auto-tuning heuristic to asses its generality (at least the one suggested by the authors). -------- The paper, despite being written nicely, suffer of some major lacks that makes it not acceptable for NIPS. In the following the issues found in the paper: 1) In Eq(2) when constructing the adjecency matrix, the ranges of the distances d(...) and \delta(...) are the same ? otherwise is not possible to merge them into the same function otherwise undesired effects in the final affinity may occur. 2) In line 125, d(u,v) is the same as the one in Eq(2) and line 114 ? In the line 114 d(s) is a measure of heterogeneity, in line 125 of distance and in Eq(2) of color distance. 3) Is not clear why the authors used clustering method that requires an input parameter K when other clustering algorithm exists and that do not require such parameter (i.e. dbscan, affinity propagation) 4) Since the pipeline uses the spectral clusterig algorithm, why the authors did not use the Spectral Gap heuristic to determine the K ? 5) In which sense the parameter K (line 143) has been manually estimated ? did you run the algorithm several time and use the K that produces the better results ? 6) Is not written which function has been used to compute d(s) (line 114). Authors just wrote an example of a possible function to be used but not the actual one. 7) In line 114, d(s) is the same as the one used in Eq(2) ? 8) Is completely arbitrary the number of "reliable" and "unreliable" neighbors because is not explained why the authors decided to use 0.5 and 0.05 (line 120 and 122). 9) The two threshold t_s and t_d how has been set ? is not written in the text. 10) In line 132, I suppose, that the equation e^-{\alpha d^2_{ij}}... \times (0.1 + 0.9 ...) is referred to the Eq.(2). If one expand that equation it results in 0.1e^{-\alpha d_{ij} +0.9e^-{\alpha d_{ij}}e^{\beta \delta{ij}} which is actually not consistent with Eq(2). Please explain. Qualitatively the performances (in particular in Fig 3) looks very nice but there are too many details that are not clear. Grammatically speaking, the paper is written nicely withour major nor minor typos. In the following the motivations for the scores: Technical quality 1: there are too many unclear things and math formulations that are ambiguous or wrong Novelty/originality 2: because the work combine known methods from image processing and machine learning Potential impact or usefulness 2: this work can have an important impact in the society to understand the shape of neurons but due to the issues in the methodological and technical parts in am not sure that it can be applied on different dataset than the one used in this paper. It seems to be higly customized. Clarity and presentation 2: because the usage of the math notation is not correct and ambiguous.

Confidence in this Review

2-Confident (read it all; understood it all reasonably well)